# EPR Spectroscopy as a Tool to Characterize the Maturity Degree of Humic Acids

**DOI:** 10.3390/ma14123410

**Published:** 2021-06-20

**Authors:** Bozena Debska, Ewa Spychaj-Fabisiak, Wiesław Szulc, Renata Gaj, Magdalena Banach-Szott

**Affiliations:** 1Department of Biogeochemistry and Soil Sciences, University of Science and Technology, 6 Bernardynska St., 85-029 Bydgoszcz, Poland; debska@utp.edu.pl (B.D.); fabisiak@utp.edu.pl (E.S.-F.); 2Division of Agricultural and Environmental Chemistry, Warsaw University and Life Sciences—SGGW, 159 Nowoursynowska St., 02-776 Warsaw, Poland; wieslaw_szulc@sggw.edu.pl; 3Department of Agricultural Chemistry and Environmental Biogeochemistry, Poznań University and Life Sciences, 38/42 Wojska Polskiego St., 60-625 Poznan, Poland; renata.gaj@up.poznan.pl

**Keywords:** humic acids, forest soil, plant material of wheat, EPR, fluorescence spectroscopy

## Abstract

The major indicator of soil fertility and productivity are humic acids (HAs) arising from decomposition of organic matter. The structure and properties of HAs depend, among others climate factors, on soil and anthropogenic factors, i.e., methods of soil management. The purpose of the research undertaken in this paper is to study humic acids resulting from the decomposition of crop residues of wheat (*Triticum aestivum* L.) and plant material of thuja (*Thuja plicata* D.Don.ex. Lamb) using electron paramagnetic resonance (EPR) spectroscopy. In the present paper, we report EPR studies carried out on two types of HAs extracted from forest soil and incubated samples of plant material (mixture of wheat straw and roots), both without soil and mixed with soil. EPR signals obtained from these samples were subjected to numerical analysis, which showed that the EPR spectra of each sample could be deconvoluted into Lorentzian and Gaussian components. It can be shown that the origin of HAs has a significant impact on the parameters of their EPR spectra. The parameters of EPR spectra of humic acids depend strongly on their origin. The HA samples isolated from forest soils are characterized by higher spin concentration and lower peak-to-peak width of EPR spectra in comparison to those of HAs incubated from plant material.

## 1. Introduction

Humic acids (HAs) are among the most abundant organic compounds found in nature and are the main component of organic matter. These compounds are involved in all soil processes and influence its physical, chemical, and biological properties. They are the main link in carbon sequestration and release of CO_2_ into the atmosphere as well as environmental detoxification. The HAs properties of soils are determined by both habitat factors (climate, soil type, granulometric composition) and anthropogenic factors related to the way it is used. The plant material that undergoes the humification process is an important element in shaping the properties of humic acids. The directions of transformation and the intensity of the humification process depend on the properties (chemical composition) of plant materials [1,2,3]. In turn, the course of the humification process determines the properties (structure) of the resulting humic acids. The complex structure of HAs and its high variability make it difficult to characterize. Electron paramagnetic resonance (EPR) is one of the few methods that can provide structural information without artifacts or restrictive experimental conditions [4]. During humification that occurs in soils [4], semiquinone radicals are generated, among others, due to polymerization or depolymerization processes. In the process of lignin (one of the main components of plant residues) oxidation, the carboxylic or aliphatic groups are lost and, this way, most of the radicals are generated, which are, in general, paramagnetic.

The structure of organic systems such as natural organic material (NOM) entrapping free radicals can be very complex and can have influence on the EPR spectra (line shape and intensity) of the entrapped radicals. The EPR spectrum of free radicals in general consist of single lines characterized by their Landé g-factor in the range of 2.0028–2.0055 (characterizing the resonance position), shapes, peak-to-peak linewidth (Δ*B_pp_*), and intensity [5,6]. The intensity and Landé g-factor are sensitive to the chemical bonds entrapping the free electron, while shape (mixture of Lorentzian and Gaussian lines) and width of EPR lines are affected by the HAs structure of the entrapping organic system.

The properties, including their EPR spectra, of HAs extracted from the soil will depend on the soil itself, along with climate and anthropogenic factors (methods of soil management) [2,3,7,8,9]. The forest stand species composition is the main factor determining the HAs properties extracted from this type of soil [1], but in arable soils, the essential role is played by the post-harvest residue left after crop harvesting [2].

This paper is devoted to the EPR studies of humic acids extracted from the forest soils and incubated plant material both without and mixed with arable soils, which means that the studied HAs have very different origins. We will show the usefulness of EPR spectroscopy in studying the degree of “maturity” of humic acids depending on their origin.

## 2. Materials and Methods

### 2.1. Materials

In the present study, two types of humic acids were analyzed. The HAs (symbols RT1_1 and RT1_3) obtained from samples of forest soils collected around thuja trees (*Thuja plicata* D.Don.ex. Lamb). The forest soil was sampled from the area of the Forest Experiment Department (FED), Warsaw Agricultural University (WAU), at Rogow. FED WAU is located in the north-eastern part of the Lodz Uplands, between 51°45′ and 51°55′ of northern altitude and between 19°50′ and 20°10′ of eastern longitude. The HAs were isolated from soil samples that were taken from two horizons (organic horizon)—sample marked RT1_1—and mineral horizon—sample marked RT1_3. Soil samples were collected in triplicate, dried in room temperature, and sieved (2 mm). Extraction of humic acids was performed for averaged samples.

The humic acids isolated from incubated samples of plant material of wheat (*Triticum aestivum* L.) (mixture of straw and roots)—G_0—and soil mixed with plant material of wheat (mixture of straw and roots)—F_1. The incubation experiment was carried out in 5 dm^3^ plastic vials with a perforated bottom lined with 0.8 mm cellophane and a filter paper. The plant material was initially ground (2–3 mm) and homogenized to a fine powder (roots and aerial parts ratio of 1:4). The soil with leftovers was mixed in a ratio of 10:1. Incubation was carried out at 25 °C, humidity 60%, and incubation time of 10 days. The incubation process was carried out in triplicate, and humic acid extraction was performed for averaged samples.

### 2.2. Methods

#### 2.2.1. Sample Preparation

Humic acids were extracted and purified according to standard methods using the following procedure:Decalcification (24 h) with 0.05 M HCl. After centrifugation, the residue was washed with distilled water until neutralExtraction (24 h) of the remaining solid with 0.5 M NaOH, with occasional mixing, followed by centrifugationPrecipitation (24 h) of humic acids from the resulting alkaline extract with 2 M HCl to pH = 2 and centrifugationPurification of the resulting humic acids as follows: The humic acid residue was treated with a mixture of HCl/HF (950 mL H_2_O, 5 mL HCl, 5 mL HF) over a HAs residue, and was treated with distilled water until a zero reaction to chloride was achieved, then freeze-dried.

The HAs sample was lyophilized and powdered in agate mortar. Ash content in the humic acids was lower than 2%.

#### 2.2.2. Elemental Composition

The elemental composition of humic acids were analyzed using Perkin Elmer Series II 2400 CHN analyzer, (PerkinElmer, Inc. Waltham, MA, USA) [2].

#### 2.2.3. EPR Measurements

EPR spectra were recorded using X-band Varian EPR spectrometer (Varian Inc Palo Alto, GA, USA) operating at 9.45 GHz at room temperature, and operating conditions were selected to avoid power saturation and modulation broadening. External magnetic field intensities and frequencies were measured separately to ensure accurate g-values.

The HAs samples were placed in quartz tubes of 3 mm in diameter. In order to avoid the so-called “oxygen effect”, the HAs samples placed in quartz tubes were evacuated to the level of 10^−6^ Tr and then introduced to the EPR cavity. EPR signals were recorded using microwave power in the range of 1–100 mW.

The 2,2-diphenyl-1-picrylhydrazyl (DPPH) was employed as the standard to determine spins concentration.

#### 2.2.4. EPR Line Deconvolution

EPR spectra were simulated as a linear combination of pure Lorentzian and pure Gaussian distributions with “home developed” software. The deconvolution procedure is based on nonlinear best fit based on the Marquardt method [10].

The deconvolution (simulation) allowed us to determine the number of ‘pure’ lines corresponding to the g-range of organic matter present in each global spectrum. Each pure line is characterized by its g-factor and peak-to-peak linewidth.

#### 2.2.5. Fluorescence Measurements

Fluorescence spectra of HAs were recorded using an Edinburg FS5 Spectrofluorometer (Edinburg Instruments, Livingston, UK). The 340 nm light wavelength was used for excitation.

## 3. Results and Discussion

### 3.1. Elemental Composition of Humic Acids

The elemental composition is considered one of the basic features of humic acids commonly used to identify them and to draw conclusions about their structure. The key elements being part of HAs are carbon, hydrogen, oxygen, and nitrogen. The numerical values of atomic ratios (O/C, O/H) and degree of internal oxidation ω, are indicators of the oxidation state of humic acids molecules and facilitate an approximation of the degree of “maturity” [2,3,11,12]. As can be seen from the data presented in Table 1, HAs of forest soil (RT_1, RT_3) were characterized by a higher oxygen content and lower hydrogen content and, consequently, higher O/C and O/H ratios and a parameter determining the degree of internal oxidation ω, in comparison with HAs of the incubated material (G_0, F_1). In light of the given results, it may be valid to speak of a higher degree of ’maturity’ of HAs of forest soils (higher values of the O/H, O/C, and ω) in comparison with HAs isolated with samples of plant material and soil samples mixed with plant material [2,3,11,12].

### 3.2. EPR Results—General Consideration

Examples of EPR spectra of both types of humic acids, i.e., those extracted from forest soils (RT_1, RT_3) and isolated from incubated samples of plant material (G_0) and plant material mixed with soil (F_1) are presented in Figure 1. The difference in the intensities of both signals results from the difference in the concentration of paramagnetic radicals (samples of the same mass were used to record the EPR spectra). For a detailed analysis of the EPR spectra for both HAs type, three repetitions were performed for each sample. The figure represents an example of a single EPR spectra of soils. Typical ESR spectra consist of multiplets arising from the superposition of individual lines, which are Gaussian or Lorentzian in shape. Dipolar broadening in solids is characterized by Gaussian shapes, and exchange narrowing causes lines to be Lorentzian [13]. In general, both mechanisms, i.e., the dipole-dipole interaction and the exchange effect, occur simultaneously if the EPR spectrum is Lorentzian (homogeneous broadening) in the central part and has Gaussian character in the wings (inhomogeneous broadening).

The very inhomogeneously broadened system is well described by the spin-packet” approach [14]. In these formulations, each spin packet is characterized as a (homogeneously broadened) spin or group of spins with the same resonant frequency. The net inhomogeneously broadened response is then obtained as a convolution of the individual packet responses with a function describing the inhomogeneous distribution of the packet resonant frequencies due to the local fields.

### 3.3. The Procedure of ESR Line Deconvolution

Typical ESR spectra consist of multiplets arising from the superposition of individual lines which are Gaussian or Lorentzian in shape. Dipolar broadening in solids is characterized by Gaussian shapes (see arrows in Figure 2), and exchange narrowing causes the lines to be Lorentzian [14].

In order to determine the contributions of both of these mechanisms (homogeneous and inhomogeneous broadening) to the EPR spectrum, they were deconvoluted to the Lorentzian and Gaussian components. The corresponding derivatives are described by the following Equations (1) and (2):

a/Gaussian:(1)$$Ỳ\left(B\right)=e^{1/2}{Ỳ}_{m}\left(\frac{B-B_{0}}{1/2\Delta B_{pp}}\right)exp\left[-1/2{\left(\frac{B-B_{0}}{1/2\Delta B_{pp}}\right)}^{2}\right]$$

b/Lorentzian:(2)$$Ỳ\left(B\right)=\frac{16{Ỳ}_{m}\left(\frac{B-B_{0}}{1/2\Delta B_{pp}}\right)}{{\left[3+{\left(\frac{B-B_{0}}{1/2\Delta B_{pp}}\right)}^{2}\right]}^{2}}$$
where:

$\Delta B_{pp}$-peak-to-peak line width, ${Ỳ}_{m}$-amplitude of EPR spectrum, $B_{0}$-the field value where Ỳ(B) = 0.

Using the above equations, the EPR spectra were numerically deconvoluted and the results are presented in Figure 2. The accuracy of the numerical fitting in this case is better than 0.99 correlation coefficient.

The parameters of EPR spectra are summarized in the Table 2. The concentration of unpaired spins (*N_s_*) in HAs was calculated from the formula:(3)$$N_{s}=N_{DPPH}\left[I_{s}{\left(\Delta B_{s}\right)}^{2}\right]/\left[I_{DPPH}{\left(\Delta B_{DPPH}\right)}^{2}\right]$$
where:

$N_{DPPH}$ is spin concentration in the standard and $I_{s},I_{DPPH}$ is amplitudes of EPR lines for sample and standard, respectively; $\Delta B_{s},\Delta B_{DPPH}$ is the peak-to peak EPR line width.

The values of experimental parameters of EPR spectra (Table 2) are in agreement with those observed by Watanabe et al. [15] for humic acids extracted from a wide range of soil types.

Denoting the integral intensities of Gaussian and Lorentzian components of an ESR line by A_G_ and A_L_, respectively, the ratios A_G_/A_L_ were calculated and shown in Table 2. The saturation effect, i.e., the dependence of EPR spectrum on the microwave power was studied separately for each component of EPR spectrum, and the results are presented in Figure 3.

### 3.4. The Interpretation of EPR Results

It was observed that, with increasing microwave power, the ESR spectrum exhibits saturation behavior, i.e., at the highest microwave power, the Lorentzian component shows saturation effect: the line width increases and EPR line intensity decreases (Figure 3). The Gaussian component does not show a saturation effect, i.e., the intensity of the ESR signal increases proportionally to the microwave power and the peak-to-peak width of the ESR line remains almost constant (Figure 3).

According to many authors [16,17,18], we can put forward the hypothesis that the Lorentzian line shape arises from exchange narrowing and the fact that the reduction in the linewidth increases is associated with the greater delocalization of the unpaired electrons as the cluster size increases.

According to this hypothesis, we can think that humic acids isolated from forest soils are characterized by a higher degree of ‘maturity’, i.e., bigger clusters are created, leading to increased overlap of the wave functions brought about by the greater delocalization and stronger exchange narrowing, which is reflected in the value of Δ*B_pp_* of the Lorentzian component.

It should be noted that dipole-dipole interaction (inhomogeneous broadening), as indicated by Δ*B_pp_* of Gaussian components, is similar for both types of humic acids. Trubetskaya et al. [19] have shown that independently of HAs source, high molecular size fractions are weakly fluorescent. The main fluorophores, especially those emitting at long wavelength (above 500 nm), are contained in the polar and low molecular size fractions. In these longer wavelength emissions, aromatic structures bearing carboxylate and OH substituents may be involved. It seems that the above statement is confirmed by fluorescence spectra of humic acids examined in this study, as presented in Figure 4.

As it is observed in Figure 4, the fluorescence spectrum of HAs extracted from forest soils is much less intense, indicating a lower concentration of low molecular size fractions in HAs.

Regardless of the HAs type, each of the fluorescence spectra can be deconvoluted into two Gaussian bands in similar positions, i.e., peaked at (420 ± 5) nm and (520 ± 5) nm, respectively. However, these spectra differ significantly in half widths of their Gaussian components, which are, respectively: (101 ± 3) nm and (158 ± 3) nm for HAs extracted from incubated samples of plant material, and (87 ± 3) nm and (116 ± 3) nm for HAs extracted from forest soils. These can be associated with different degrees of humification.

## 4. Conclusions

The HAs samples isolated from forest soils are characterized by higher spin concentration and lower EPR spectrum in comparison to HAs isolated from incubated plant material without soil and with soil. The relationships obtained in this study showed that EPR spectroscopy is suitable for determining the properties of humic acids of various origins.

EPR spectra can be divided into Gaussian and Lorentzian components, which exhibit different saturation properties, proving the existence of two sub-structures in HAs. This reduction in linewidth Δ*B_pp_* may be attributed to motional or exchange narrowing, which is related to increased wavefunction overlap due to increased size of sp^2^ clusters (higher degree of ‘maturity’ of HAs) [20].

This implies that the unpaired spins are likely to be associated more with π-type rather than σ-type radicals [21]. The above hypothesis seems to be confirmed by the corresponding fluorescence spectra, which provides the opportunity to distinguish between soil types by using EPR spectra.

## Figures and Tables

**Figure 1 materials-14-03410-f001:**
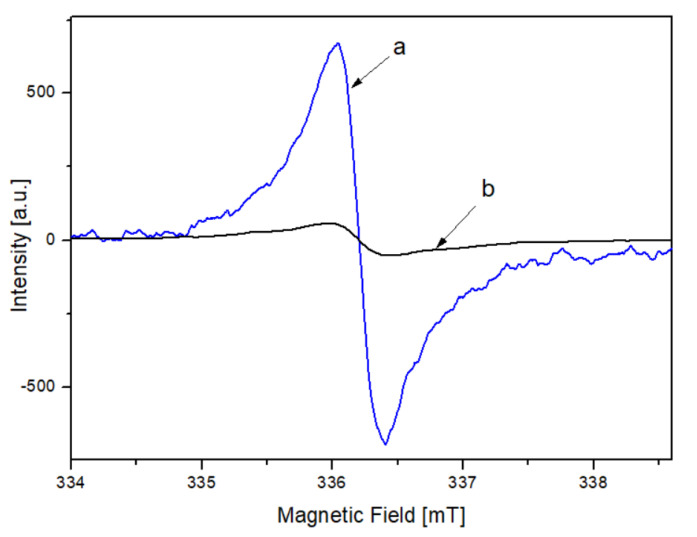
EPR spectra of the HAs extracted from forest soils (**a**) and isolated from incubated samples of plant material (**b**).

**Figure 2 materials-14-03410-f002:**
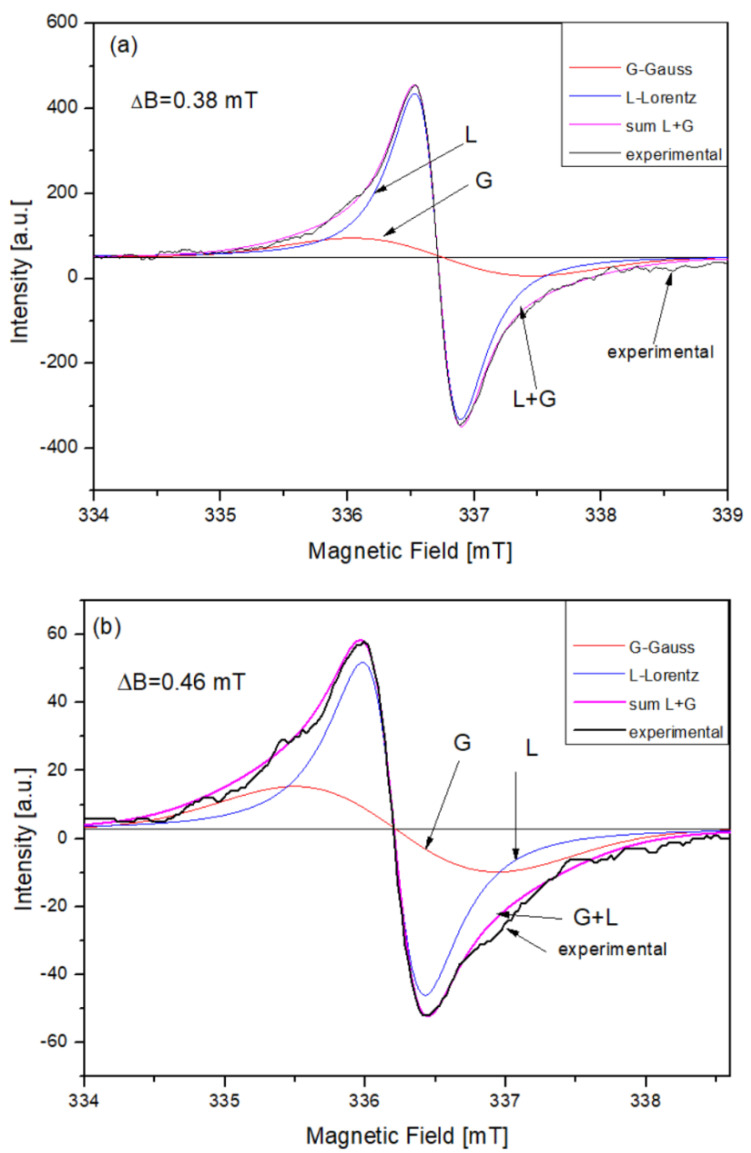
The deconvoluted EPR spectra recorded for samples: (**a**) extracted from forest soils and (**b**) isolated from incubated samples of plant material. Δ*B*—peak-to-peak width of experimental line. G—Gaussian; L—Lorentzian components; G + L—sum of both components. (The accuracy of the numerical fitting in this case is better than 0.99 correlation coefficient).

**Figure 3 materials-14-03410-f003:**
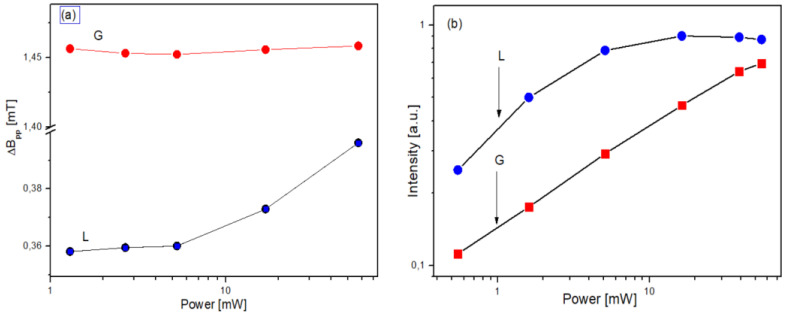
The parameters of EPR spectrum from Figure 2a as a function of microwave power: (**a**)/ line widths of Gaussian and Lorentzian components $\Delta B_{G\;}$, and $\Delta B_{L}$ (**b**)/ A_G_ and A_L_ of Gaussian and Lorentzian components, respectively.

**Figure 4 materials-14-03410-f004:**
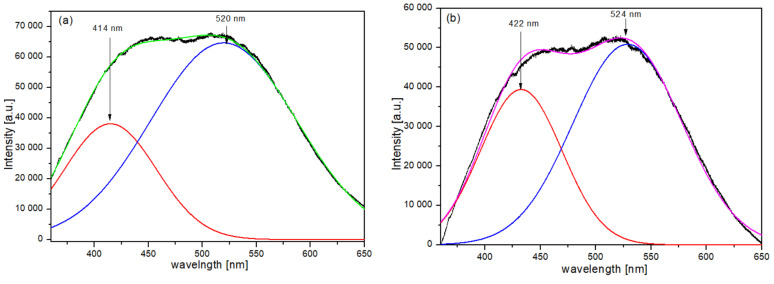
Florescence spectra (excitation 340 nm) of the HAs isolated from: (**a**) incubated samples of plant material, (**b**) forest soils.

**Table 1 materials-14-03410-t001:** Elemental composition of the humic acids (in atomic percentage).

Sample	C	H	N	O	O/C	O/H	ω *
RT_1	35.58	44.20	1.12	19.10	0.537	0.432	−0.074
RT_3	33.22	44.88	1.93	19.97	0.601	0.445	0.025
G_0	36.31	48.36	1.10	14.23	0.392	0.294	−0.457
F_1	37.28	45.27	1.61	15.84	0.425	0.350	−0.235

*—internal oxidation degree (ω = (2O + 3N-H)/C).

**Table 2 materials-14-03410-t002:** The parameters of the EPR lines (average values of EPR spectra of the measured values along with standard deviations).

HAs Extracted from Forest Soils		HAs Isolated from Incubated Samples of Plant Material	
N_s_ (spin/g)	(3.24 × 10^18^ ± 2%)	N_s_ (spin/g)	(4.72 × 10^17^ ± 2%)
Δ*B_pp_*-experimental	(0.38 ± 0.005) mT	Δ*B_pp_*-experimental	(0.46 ± 0.005) mT
Δ*B_pp_*-Lorentzian	(0.355 ± 0.002) mT	Δ*B_pp_*-Lorentzian	(0.45 ± 0.002) mT
Δ*B_pp_*-Gaussian	(1.52 ± 0.002) mT	Δ*B_pp_*-Gaussian	(1.505 ± 0.002) mT
g-value	2.0024 ± 0.0002	g-value	2.0028 ± 0.0002
A_G_/A_L_-ratio	2.47 ± 2%	A_G_/A_L_ -ratio	3.71 ± 2%

## Data Availability

The data presented in this study are available on request from the corresponding author.

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
