# Peer review of "EPR Spectroscopy as a Tool to Characterize the Maturity Degree of Humic Acids"

_materials, 2021, doi:10.3390/ma14123410_

Round 1

Reviewer 1 Report

The authors conducted a study to determine if EPR can be used to detect differences in soil humus, based on the idea that differences in humus would be linked to differences in plant substrate sources. They took four total samples from a forest, with two levels of “organic,”  and four samples from incubations with wheat under two different treatments. The premise of this study is sound and interesting, but the paper lacks from replication and being difficult to read, given some disparate organization and presentation of new methods in the results/discussion section.

I recommend that they authors use complete paragraphs in the methods, instead of bullet points. They would need to also add in a study area section to better describe the source of the soils

Many abbreviations are mentioned before or without them being defined, such as how EPR is first given in the abstract, but later defined, and both thuja and wheat need to have scientific names added to the abstract. 

I recommend that the authors use complete paragraphs in the introduction, probably pulling in some of the more complete descriptions of humus chemistry from the discussion. As it is, the introduction feels like a few disparate sentences, with no clear connections. Then, the authors should add in a quick summary of their experimental approach and hypothesis to end the introduction, tying together the information presented therein

The figures are disorganized and the captions are incomplete, so they were difficult to review properly. For instance, figure 2 does not have an “a” panel included, and figure 1 seems to be averaged over treatments, but It’s unclear why the raw data wasn’t shown in figure 1. 

I’m concerned that there were was not strong replication of soil sources, so it is difficult to determine if the soil itself is just different or if the reason for the differences in EPR are from the plant substrates. Also, it’s difficult to tell whether the main take away is that this is a demonstration of a method (EPR) or an evaluation of EPR signals for these specific plant types. 

Author Response

Thank you very much for your insightful review and valuable suggestions and comments. We tried to make appropriate changes in the corrected manuscript as requested by the reviewer.

The chapter Introduction was revised and appriopriate changes were introduced. Materials and Methods and Results and Discussion chapters have been ordered as requested by the Reviewer.

Figure 2 has been supplemented with the missing panel. Fig.1 contains raw experimental data (EPR spectra cannot be averaged, only smothing can be made). Note: In all drawings, lines marked experimental represent lines on the raw data.

In general, it is considered that if the EPR spectra have the same peak-to-peak width, the same g-factor and the same spin concentration, it is a characteristic feature of the studied  material.

The main assumption of the presented work was to show that performing a numerical analysis of EPR spectra allows to detect subtle differences in HA's structures resulting from their origin and differences in humification processes. In our understanding, further systematic research on this topic is needed.

All introduced changes are appropriately marked in the revised manuscript.

See the detailed changes below:

ABSTRACT:                        

There was: The major indicator of soil fertility and productivity are humic acids (HAs) arise from degradation of organic matter.

There is: The major indicator of soil fertility and productivity are humic acids (HAs) arise from decomposition of organic matter.

There was: The purpose of the research undertaken in this work is to study humic acids resulting from the decomposition of crop residues of wheat and plant material of thuja using EPR spectroscopy.

In the present work we report Electron Paramagnetic Resonance studies carried out on two types of HAs extracted from forest soil and incubated samples of plant material (mixture of wheat straw and roots).

There is: The purpose of the research undertaken in this paper is to study humic acids resulting from the decomposition of crop residues of wheat (Triticum aestivum L.) and plant material of thuja (Thuja plicata D.Don.ex. Lamb) using Electron Paramagnetic Resonance (EPR) spectroscopy.

In the present paper we report EPR studies carried out on two types of HAs extracted from forest soil and incubated samples of plant material (mixture of wheat straw and roots) without soil and mixed with soil.

There was: The HAs samples isolated from forest soils are characterized by higher spin concentration and lower peak-to-peak width of EPR spectrum in comparison to HAs incubated from crop plant material.

There is: The HAs samples isolated from forest soils are characterized by higher spin concentration and lower peak-to-peak width of EPR spectrum in comparison to HAs incubated from plant material.

INTRODUCTION:

In the introduction section the following has been removed:

The quantitative and qualitative composition of soil organic matter is determined both by habitat factors (climate, soil type, grain size composition) and by anthropogenic factors related to the way it is used. Humus substances, which occur in mineral soils in small amounts, due to their specific nature, often play a significant role in the course of many physical, chemical and biochemical processes in the soil profile [1-3]

and added:

Humic acids (HAs) are among the most abundant organic compounds found in nature and are the main component of organic matter. These compounds are involved in all soil processes and influence its physical, chemical and biological properties. They are the main link in carbon sequestration and release of CO2 into the atmosphere as well as environmental detoxification. The HAs properties of soils are determined by both habitat factors (climate, soil type, granulometric composition) and anthropogenic factors related to the way it is used. Plant material that undergoes the humification process is an important element in shaping the properties of humic acids. The directions of transformation and the intensity of the humification process depend on the properties (chemical composition) of plant materials [1-3]. In turn, the course of the humification process determines the properties (structure) of the resulting humic acids. The complex structure of HAs and its high variability make it difficult to characterize HAs. Electron Paramagnetic Resonance (EPR) is one of the few methods that can provide structural information without artifacts or restrictive experimental conditions [4].

It has been removed:

The EPR spectra of the free radicals present in HAs are characterized by g-factors having values in the range of 2.0028 - 2.0055 and single EPR line with peak-to-peak linewidth < 0.7 mT [6].

The purpose of the study has also been improved:

There was: This work is devoted to EPR the studies of humic acids extracted from the forest soils and arable soils what means that the studied HAs have very different origin. Our goal at this stage of research is to show the possibilities and usability of EPR spectroscopy in studying the degree of humification of humic acids depending on their origin.

There is: This paper is devoted to the EPR studies of humic acids extracted from the forest soils and incubated plant material without and mixed with arable soils what means that the studied HAs have very different origin. We will show the usefulness of EPR spectroscopy in studying the degree of “maturity” of humic acids depending on their origin.

MATERIALS AND METHODS

2.1. Materials

There was:

In the present study two types of HAs were analyzed:

  • humic acids (HAs – symbols RT1_1 and RT1_3) obtained from samples of forest soils collected around thui trees (Thuja plicataDon.ex. Lamb), located in the Forest Experimental Station of SGGW in Rogow. HAs were taken from two levels (organic level) - sample marked RT1_1 and mineral level - sample marked RT1_3, in two independent samples;
  • isolated from incubated samples of plant material of wheat (Triticum aestivum) (mixture of straw and roots) - G_0 and soil mixed with plant residues of wheat - F_1, from two independent samples each. The incubation experiment was carried out in 5 dm3 plastic vials with a perforated bottom lined with 0.8 mm cellophane and a filter paper. The plant material was initially ground (2-3 mm) and homogenized to a fine powder (roots and aerial parts ratio of 1:4). The soil with leftovers was mixed in a ratio of 10:1. Incubation was carried out at 250C, humidity 60%, incubation time 10 days.

There is:

In the present study two types of humic acids  were analyzed. The HAs (symbols RT1_1 and RT1_3) obtained from samples of forest soils collected around thui trees (Thuja plicata D.Don.ex. Lamb). The forest soil sampled from the area of the Forest Experiment Department (FED), Warsaw Agricultural University (WAU), at Rogow. FED WAU is located in the north-eastern part of the Lodz Uplands, between 51°45’ and 51°55’ of northern altitude and between 19°50’ and 20°10’ of eastern longitude. The HAs were isolated from soil samples that were taken from two horizons (organic horizon) - sample marked RT1_1 and mineral horizon - sample marked RT1_3. Soil samples were collected in triplicate, dried in room temperature, and sieved (2 mm). Extraction of humic acids was performed for averaged samples.

The humic acids isolated from incubated samples of plant material of wheat (Triticum aestivum L.) (mixture of straw and roots) - G_0 and soil mixed with plant material of wheat (mixture of straw and roots) - F_1. The incubation experiment was carried out in 5 dm3 plastic vials with a perforated bottom lined with 0.8 mm cellophane and a filter paper. The plant material was initially ground (2-3 mm) and homogenized to a fine powder (roots and aerial parts ratio of 1:4). The soil with leftovers was mixed in a ratio of 10:1. Incubation was carried out at 250C, humidity 60%, incubation time 10 days. The incubation process was carried out in triplicate, humic acid extraction was performed for averaged samples.

2.2. Methods

 subsections added:   2.2.1. Sample preparation;  2.2.2. Elemental composition; 2.2.3. EPR measurements; 2.2.4. EPR line deconvolution; 2.2.5. Fluorescence measurements

There was:

EPR spectra were recorded using X-band Varian EPR spectrometer operating at 9.45 GHz at room temperature, following the procedure based on nonlinear best fit based on the Marquardt method [10]. EPR spectra were recorded at room temperature and operating conditions were selected to avoid power saturation and modulation broadening. External magnetic field intensities and frequencies were measured separately to ensure accurate g-values.

The 2,2-diphenyl-1-picrylhydrazyl (DPPH) was employed as the standard to determine spins concentration. EPR spectra were simulated as a linear combination of pure Lorentzian and pure Gaussian distributions with ‘home developed’ software.

The simulation allowed us to determine the number of ‘pure’ lines corresponding to the g-range of organic matter present in each global spectrum. Each pure line is characterized by its g-factor and peak-to-peak linewidth. The HAs samples were placed in quartz tubes of 3 mm in diameter. In order to avoid the so-called “oxygen effect” the HAs samples placed in quartz tubes were evacuated to the level of 10-6Tr and then introduced to the EPR cavity. EPR signals were recorded using microwave power in the range of 1-100mW.

Fluorescence spectra of HAs were recorded using an Edinburg FS5 Spectrofluorometer.

There is:

2.2.3.EPR measurements

EPR spectra were recorded using X-band Varian EPR spectrometer operating at 9.45 GHz at room temperature and operating conditions were selected to avoid power saturation and modulation broadening. External magnetic field intensities and frequencies were measured separately to ensure accurate g-values.

The HAs samples were placed in quartz tubes of 3 mm in diameter. In order to avoid the so-called “oxygen effect” the HAs samples placed in quartz tubes were evacuated to the level of 10-6Tr and then introduced to the EPR cavity. EPR signals were recorded using microwave power in the range of 1-100mW.

The 2,2-diphenyl-1-picrylhydrazyl (DPPH) was employed as the standard to determine spins concentration.

2.2.4 EPR line deconvolution

EPR spectra were simulated as a linear combination of pure Lorentzian and pure Gaussian distributions with ‘home developed’ software. The deconvolution procedure is based on nonlinear best fit based on the Marquardt method [10].

The deconvolution (simulation) allowed us to determine the number of ‘pure’ lines corresponding to the g-range of organic matter present in each global spectrum. Each pure line is characterized by its g-factor and peak-to-peak linewidth.

  • Fluorescence measurements

Fluorescence spectra of HAs were recorded using an Edinburg FS5 Spectrofluorometer. The 340 nm light wave length was used for excitation.

RESULTS AND DISCUSSION

subsections added:  3.1. Elemental composition of humic acids; 3.2. EPR results – general consideration; 3.3. The procedure of ESR line deconvolution; 3.4. The interpretation of EPR results.

The titles of Tables 1 and 2 have been corrected, Table 1 has been moved to the Results and Discussion section.

In the results and discussion section, the section on EPR measurements, EPR line deconvolution and Fluorescence measurements has been edited.

For example in the subsection  3.2. EPR results – general consideration was added: …. samples of plant material (G_0) and plant material mixed with soil  (F_1) are presented in Figure 1. The difference in the intensities of both signals results from the difference in the concentration of paramagnetic radicals (samples of the same mass were used to record the EPR spectra).

Figure 2 has been supplemented with the missing panel. Fig.1 contains raw experimental data (EPR spectra cannot be averaged, only smothing can be made). Note: In all drawings, lines marked experimental represent lines on the raw data.

 CONCLUSSION

Was added…..HAs isolated from incubated plant material without soil and with soil. The relationships obtained in this study showed that EPR spectroscopy is suitable for determining the properties of humic acids of various origins.

All introduced changes are appropriately marked in the revised manuscript.

Reviewer 2 Report

In order to publish this manuscript, minor revision is needed and my recommendations are:

  1. minor English corrections and typos are needed
  2. the introduction chapter must be improved, by presenting the advantages offered by this  method for the characterization of humic acids
  3. Please improve the quality and resolution of the pictures

Author Response

Thank you very much for your valuable comments. Appropriate changes have been made.

For example:

In the introduction section the following has been removed:

The quantitative and qualitative composition of soil organic matter is determined both by habitat factors (climate, soil type, grain size composition) and by anthropogenic factors related to the way it is used. Humus substances, which occur in mineral soils in small amounts, due to their specific nature, often play a significant role in the course of many physical, chemical and biochemical processes in the soil profile [1-3]

and added:

Humic acids (HAs) are among the most abundant organic compounds found in nature and are the main component of organic matter. These compounds are involved in all soil processes and influence its physical, chemical and biological properties. They are the main link in carbon sequestration and release of CO2 into the atmosphere as well as environmental detoxification. The HAs properties of soils are determined by both habitat factors (climate, soil type, granulometric composition) and anthropogenic factors related to the way it is used. Plant material that undergoes the humification process is an important element in shaping the properties of humic acids. The directions of transformation and the intensity of the humification process depend on the properties (chemical composition) of plant materials [1-3]. In turn, the course of the humification process determines the properties (structure) of the resulting humic acids. The complex structure of HAs and its high variability make it difficult to characterize HAs. Electron Paramagnetic Resonance (EPR) is one of the few methods that can provide structural information without artifacts or restrictive experimental conditions [4]. It has been removed:

The EPR spectra of the free radicals present in HAs are characterized by g-factors having values in the range of 2.0028 - 2.0055 and single EPR line with peak-to-peak linewidth < 0.7 mT [6].

The purpose of the study has also been improved:

There was: This work is devoted to EPR the studies of humic acids extracted from the forest soils and arable soils what means that the studied HAs have very different origin. Our goal at this stage of research is to show the possibilities and usability of EPR spectroscopy in studying the degree of humification of humic acids depending on their origin.

There is: This paper is devoted to the EPR studies of humic acids extracted from the forest soils and incubated plant material without and mixed with arable soils what means that the studied HAs have very different origin. We will show the usefulness of EPR spectroscopy in studying the degree of “maturity” of humic acids depending on their origin.

The resolution of the drawings has been improved.

All introduced changes are appropriately marked in the revised manuscript.